# Observation of bulk quadrupole in topological heat transport

Guoqiang Xu[1,5], Xue Zhou[2,5], Shuihua Yang [1,5], Jing Wu [3,4] &
Cheng-Wei Qiu [1] ✉

The quantized bulk quadrupole moment has so far revealed a non-trivial boundary state with lower-dimensional topological edge states and in-gap zero-dimensional corner modes. In contrast to photonic implementations, state-of-the-art strategies for topological thermal metamaterials struggle to achieve such higher-order hierarchical features. This is due to the absence of quantized bulk quadrupole moments in thermal diffusion fundamentally prohibiting possible band topology expansions. Here, we report a recipe for generating quantized bulk quadrupole moments in fluid heat transport and observe the quadrupole topological phases in non-Hermitian thermal systems. Our experiments show that both the real- and imaginary-valued bands exhibit the hierarchical features of bulk, gapped edge and in-gap corner states—in stark contrast to the higher-order states observed only on real-valued bands in classical wave fields. Our findings open up unique possibilities for diffusive metamaterial engineering and establish a playground for multipolar topological physics.

Topological states of matter have found explosive developments across various classical wave fields[1–5]. In an adiabatic system, Hermiticity lies at the foundation of these emerging topological properties[6,7], as it ensures the real-valued eigenvalues and orthogonal eigenstates. When considering open systems, additional interactions with the ambient raise the non-Hermiticities. Though these dissipations fail the fundamental bulk-boundary correspondence[8,9] defined in Hermitian system, a plethora of exotic properties are empowered, such as parity-time symmetry[10–12], skin effects[13,14], as well as Weyl exceptional rings in cold atomic gas[15], photonics[16], and semimetal[17]. The newly predicted higher-order topological insulators (HOTI) have further paved an avenue toward studying hierarchical features in both Hermitian[18–23] and non-Hermitian[24–27] systems. Featuring a quantized bulk quadrupole moment[18], the Benalcazar–Bernevig–Hughes (BBH) model holds the key for realizing a minimal quadrupole topological insulator (QTI) possessing positive and negative couplings[18–20]. Moreover, a modified non-Hermitian BBH model indicates that both the on-site

non-Hermiticities[24] and the Hermiticities[25] can derive the quadrupole topological phases and modulate the higher-order transitions in real-valued bands[24–27].

It is recently found that dissipative diffusion is fundamentally governed by skew-Hermitian physics and characterized by a purely imaginary Hamiltonian[28,29]. It thus enables the counter-intuitive topological features in heat transport, such as non-Hermitian topological insulating phases[30] and Weyl exceptional rings[31]. On the other hand, even the state-of-the-art methods[28–31] fail to create non-Hermitian thermal quadrupole topological phases, due to the absent bulk quadrupole moment and undefined negative couplings in heat transfer. Therefore, to date, non-Hermitian BBH model seems not applicable to heat transport, and quadrupole topological phases in thermal diffusion are still elusive at large.

Here, we reveal the existence of quadrupole moment and non-Hermitian quadrupole topological phases in heat transport. It is essentially realized by judicious configurations of controllable thermal

[1]Department of Electrical and Computer Engineering, National University of Singapore, Kent Ridge, Singapore 117583, Singapore. [2]School of Computer Science and Information Engineering, Chongqing Technology and Business University, Chongqing 400067, China. [3]Institute of Materials Research and Engineering, Agency for Science, Technology and Research, Singapore, Singapore. [4]Department of Materials Science and Engineering, National University of Singapore, Singapore, Singapore. [5]These authors contributed equally: Guoqiang Xu, Xue Zhou, Shuihua Yang. ✉e-mail: chengwei.qiu@nus.edu.sg

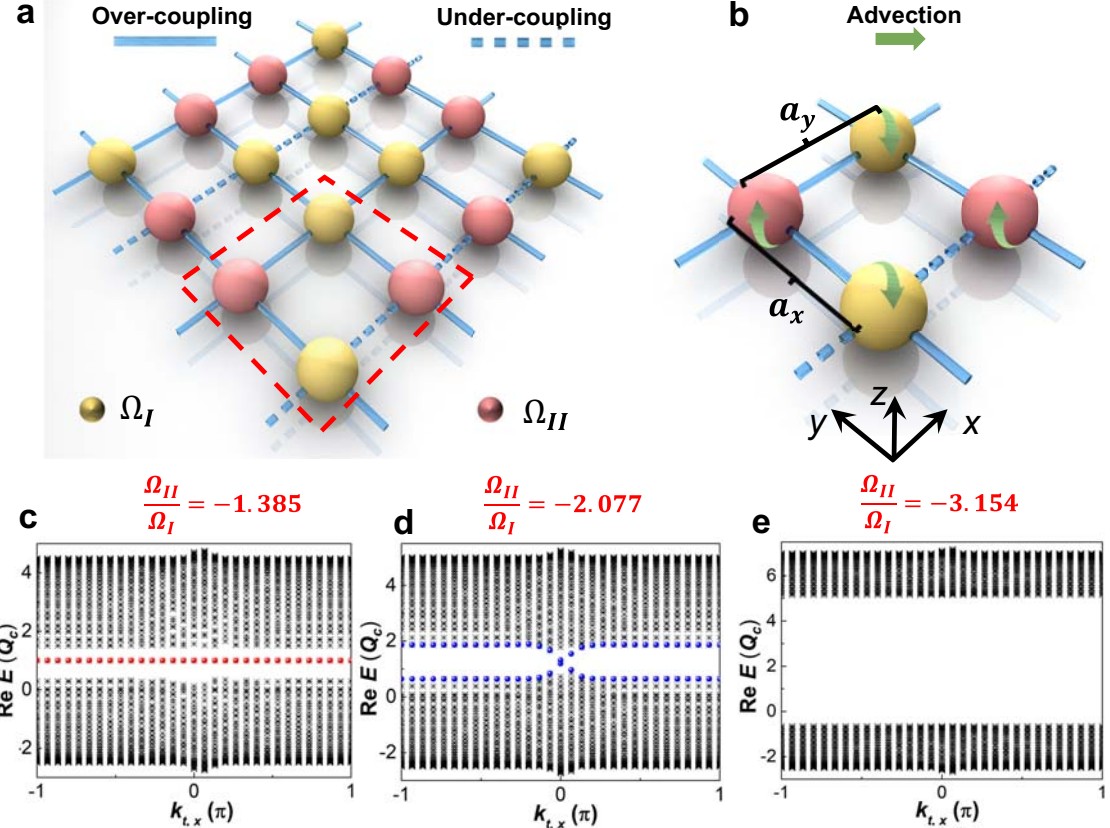

**Fig. 1 | Quadrupole topological phases in heat transport induced by Hermitian advection and relevant band structures. a** presents a square-lattice consisting of 16 sites in a grid thermal system. The red border indicates a four-site unit-structure. $\Omega_{I/II}$ represents the magnitude of the angular velocities imposed on each site. **b** Schematic unit-structure with four sites, and $a_{x/y}$ presents the widths for heat transport between the centers of neighboring sites. The light-yellow and light-red colors indicate the counter-advections imposed on corresponding sites. The green arrows present the advective directions. Each advection can be decoupled as two couplings between neighboring sites. **c**–**e** plot of the real spectra of the dispersion induced by Hermitian advection respectively at the in-gap corner, gapped edge, and trivial bulk states. The horizontal and vertical axes denote the effective Bloch wave numbers and real value of the eigenvalue. The red and blue dots in (**c** and **d**) respectively indicate the corner and edge states.

couplings between neighboring sites. In contrast to the fact that the higher-order features can only be experimentally observed on real-valued bands in classical wave fields, we capture these states on both real- and imaginary-valued bands. We then experimentally demonstrate these thermal quadrupole topological phases, and observe significant temperature localizations at the bulk, edge, and corner of the fabricated samples. Our work sheds light upon establishing quantized bulk quadrupole moments in thermal systems and unlocking rich topological phase transitions in various diffusions[32–35] and higher-order topological insulators in diffusion systems in purely thermal conduction[36,37].

## Results

### Generation of non-Hermitian quadrupole topological insulator in fluid heat transport

We first consider a convective fluid heat transport with multiple discrete sites as illustrated in Fig. 1a, b. Each site indicates a finite-volume of heat transfer process, and the grid lines between neighboring sites correspond to their thermal couplings. In stark contrasts to classical wave dynamics, the fluid heat transport is based on continuous model following conservation laws. Thus, the continuous conditions should be considered for quantization (Supplementary Note 1). We adopt tunable advections on each site to provide the necessary modulation and create effective oscillations, thus further forming an effective unit-structure consisting of four neighboring sites (Fig. 1b). Such unit-structures can be periodically configured to establish an effective 2D

square-lattice with 16 sites (Fig. 1a) in heat transfer. The general heat energy equation[38] for each site can be expressed as

$$
\frac{\partial T_{ij}}{\partial t} = \frac{\kappa}{\rho c}\nabla(\nabla T_{ij}) \pm \Omega_{I/II}R(\theta)\cdot\nabla T_{ij} + \underbrace{\sum\frac{h}{\rho c a_{x/y}}\Delta T}_{\text{int racell}} + \underbrace{\sum\frac{\beta h}{\rho c a_{x/y}}\Delta T}_{\text{int ercell}}.
$$

(1)

In Eq. (1), $\rho$, $c$, and $\kappa$ respectively denote the density, specific heat, and thermal conductivity of the site. Each site is depicted by its position ($i$ and $j$), and $T_{ij}$ denotes the corresponding temperature (Supplementary Fig. 1a). $\Omega_{I/II}$ represents the magnitude of the angular velocities of the convection imposed on each site, and $R$ and $\theta$ respectively denote the radial and azimuth components in the $x$-$y$ plane. $h$ indicates the heat transfer coefficient of the selected site, and $a_{x/y}$ presents the widths for heat transport between the centers of neighboring sites (Fig. 1b). Here we make $a_x = a_y = a$ to ensure an effective square unit-structure, and let $Q_c = \frac{h}{\rho c a}$ with the unit of $s^{-1}$ stand for the thermal coupling strength. $\beta$ is the ratio between the intercell and intracell thermal coupling strengths, and its value is 1 when the heat exchange areas of the intracell and intercell components are same (Methods). Taking into account thermal couplings in such a 2D network, two components along the $x$ and $y$ directions can be decoupled from the imposed advections on each site, i.e., $\Omega_{I/II}\cdot\cos\theta$ and $\Omega_{I/II}\cdot\sin\theta$. In that case, a diffusive analog to the quantized bulk

quadrupole moment could emerge and exhibit characteristic quadrupole fields in temperature distributions when modulating the advection and thermal coupling in the unit-structure (Supplementary Note 3). It is worth to note that such two components originate from the non-vanishing first-order drift terms of the advective vectors, thus further indicating the different forms for quantization between the governing function Eq. (1) and the well-established Schrodinger equation.

Since the intrinsic thermal coupling is governed by skew-Hermitian physics[27–30], the imposed advections act as the real Hermiticities, which are equivalent to the roles of gain and loss in photonics. Note that the neighboring sites are coupled via the heat exchanges induced by intracells and intercells. The tilted connections (Supplementary Fig. 2) between two adjacent sites result in different orientations of isotherms and coupling degrees under the same advections (Supplementary Fig. 3). Such an implementation enables the over-coupling and under-coupling, with respect to the reference coupling strength in un-tilted configurations (Supplementary Note 3). Due to the advective velocity vectors along $x$ and $y$ directions, the temperature fields are available to propagate with two components respectively along the advections like a wave. Thus, a wave-like solution $T_{ij} = A e^{i(k_x x - \omega_x t + \varphi_x + k_y y - \omega_y t + \varphi_y)}$ on each site can be adopted to reveal the oscillatory temperature field propagations, where $k_{x/y} = \frac{2\pi}{l_{x/y}} = R_{x/y}^{-1}$, $\omega_{x/y} = -i\left(\frac{\kappa \cdot k_{x/y}^2}{\rho c} + (1+\beta)Q_c\right) - \Omega_{I/II,x/y}$, $A$, and $\varphi_{x/y}$ indicate the effective wave numbers, the complex angular frequencies, the amplitude of temperature field, and the initial phase angles respectively along the $x$ and $y$ directions. The values of $\varphi_{x/y}$ are respectively 0 and $\pi$ for corresponding diagonal hot and cold sites. Then, the effective Hamiltonian of a four-site unit-structure can now be written as Supplementary Eq. 5, where the real and imaginary parts denote the two decoupled temperature field components respectively along $x$ and $y$ directions.

The complex angular frequency and eigenvalues (Supplementary Note 2) imply that both the advections and corresponding couplings result in the complex bands. The imaginary angular frequency $-i\left(\frac{\kappa \cdot k_{x/y}^2}{\rho c} + (1+\beta)Q_c\right)$ originates from the intrinsic conduction and the thermal couplings, while the real angular frequency $\Omega_{I/II}$ represents the effective momentum induced by the imposed advections towards different azimuths. These two parts simultaneously determine the amplitudes and the movements of the dynamic temperature field, thus retaining the possibilities of exciting significant hierarchical states with two distinct recipes, i.e., modulating the Hermitian advection and the non-Hermitian coupling. We then fabricate a square-lattice with 16 sites and immerse it into water (Methods). All fabricated sites are hollow in order to make water pass through and connect with tilted channels possessing tailored thermal coupling strengths. These sites are of the same size and act as advective balls to provide the needed advections.

## Non-Hermitian quadrupole topological phase induced by Hermitian advection

We first focus on the quadrupole topological phases enabled by the Hermitian advection. For example, we make $\Omega_I = 1.3 Q_c > 0$ and implement advective modulations under $|\Delta\Omega| = |\Omega_I - \Omega_{II}| \geq 2\sqrt{2}Q_c$ to ensure the real eigenvalues (Supplementary Notes 2 and 4). The real band structures of the first Brillouin zones under specific advections are presented in Supplementary Fig. 6a, which imply the topological phase transition via solely modulating the Hermitian advection and represent a class of topological quadrupole phases, embracing the in-gap 0D and gapped 1D topological modes. We then calculate the dispersion relations to further validate the existence of these quadrupole

topological phases. The robust in-gap corner state (red dots) and gapped edge state (blue dots) are respectively presented in Fig. 1c and d, revealing these higher-order states with the advective configurations of $\Omega_{II} = -1.385\Omega_I$ and $\Omega_{II} = -2.077\Omega_I$. The completely gapped bands illustrated in Fig. 1e with $\Omega_{II} = -3.154\Omega_I$ present a thermal analog of a trivial insulator.

We then fabricate a thermal system consisting of 12 sites (9 square lattices) along the $x$ and $y$ directions (Fig. 2a) to manifest these nontrivial states. In order to ensure the topological transitions solely via the Hermitian advection (Supplementary Eq. 11), the same structures are adopted in all coupling channels to retain the same intercell and intracell thermal coupling strength ($\beta = 1$). One of the imposed advections on a pair of diagonal sites in one unit-structure (Fig. 2a) is adopted as $\Omega_I = 1.3 Q_c$ based on the calculated dispersion in Fig. 1c, while we sweep advection $\Omega_{II}$ on the other pair of diagonal sites within the range of $[-3.154\Omega_I, 0]$ to search for corresponding real angular frequency (Supplementary Eq. 11). Due to the effective quantized quadrupole moment enabled by the above advective arrangements (Supplementary Notes 2–4), the eigenfrequency spectrum indicates that significant hierarchical features discretely distribute along the real-valued band and localize on one gapless imaginary-valued band (Fig. 2b). When Re$f$ respectively approaches 0 and $4.81 Q_c$, the trivial bulk states showcase the gaps between these two branches in the real-valued band. We choose three sites respectively at the center, edge, and corner of the sample (marked as a square in Fig. 2a), and capture their responses under changing Re$f$ as plotted in Fig. 2c. Here, we take the absolute values of the normalized temperatures $I = \left|\frac{(T^* - \bar{T}_{mea})}{\Delta T_{mea}}\right|$ to evaluate field intensities, where $T^*$, $\bar{T}_{mea}$ and $\Delta T_{mea}$ respectively denote the target temperature at specific measured points, the average temperature of the system, and the difference between the highest temperature and $\bar{T}_{mea}$ at the measured moments. Two peaks of the field intensities are observed at corresponding Re$f$ to the bulk branches as predicted in Fig. 2b. Similar to the responses in the bulk, the gapped edge states also exhibit two peaks as the gradient blue area in Fig. 2c. The four in-gap corner states emerge when Re$f \sim 3.05 Q_c$. In that case, only one peak is found on the field intensity distribution. To further experimentally demonstrate these quadrupole topological phases, we measure the temperature distributions at corresponding Re$f$ by modulating the advections as shown in Fig. 2d–f. The corresponding numerical results for these behaviors are shown in Supplementary Fig. 7. Note that, the observed behaviors simultaneously contain effects on multiple fields for fluid heat transport. The findings in Fig. 2 are exhibited with temperature distributions, since the energy equation of fluid transport naturally satisfy the description of systemic energies of Hamiltonian. More intuitively, the systemic velocity and pressure distributions described by momentum equation of Navier-Stokes equation can be also adopted to directly present these behaviors in the real vector space (Supplementary Note 4.4). These theoretical, numerical, and experimental findings reveal the quadrupole topological phases in real-valued bands solely induced by Hermitian advection in a thermal system. All these demonstrated fields are typical transport phenomena within the fluids. These transport quantities of energy, mass, and momentum follows the conservation laws formulated by the constitutive equations of continuity equation, momentum equation, and energy equation of the fluid heat transport. The same continuous mechanism and mathematical frameworks of the constitutive equations between these different fields enable the similar processes of conserved transport and quantization, which can be generally described by the balance among the conserved quantities entering and leaving the control volume, the additional generations of all the original components for the conserved quantities, and the non-zero accumulations (the net flows) retaining in the system after the conservation process. Such a process is characterized by

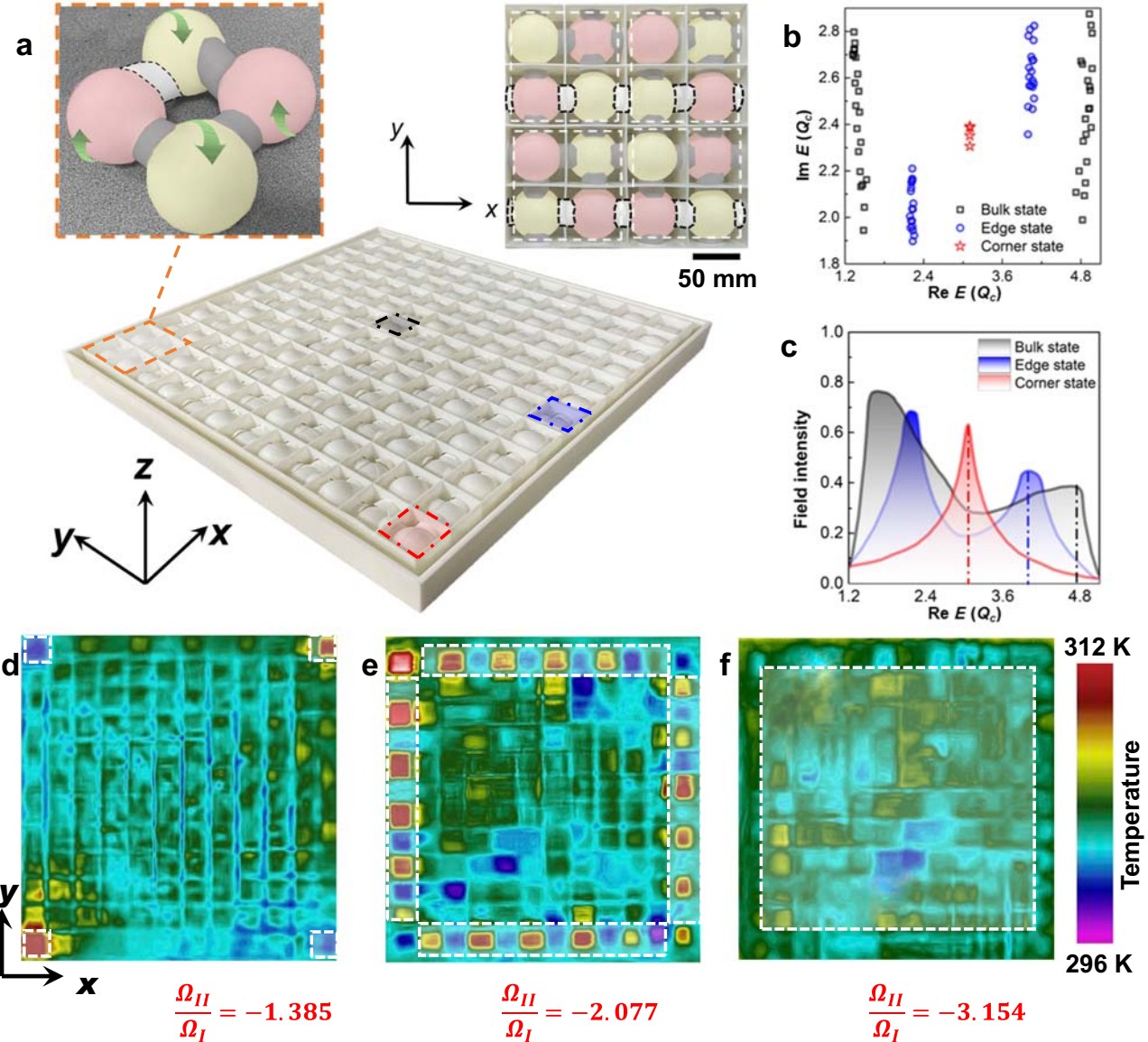

**Fig. 2 | Quadrupole topological phases in heat transport solely induced by Hermitian advection. a** Photo of a fabricated sample with 9 square-lattices made of epoxy resin, while water is fully imposed to each site as the working fluid. The left and right upper insets present the connection of a fabricated unit-structure and one square lattice with 16 sites. The grey-shadowed and black-dashed areas respectively indicate the over-coupling and under-coupling channels. The green arrows indicate the directions of the imposed advections. **b** Spectra of the thermal quadrupole topological phases. **c** Measured temperature field intensities at corresponding boundaries. The measured regions are marked by colored borders in (**a**). **d**–**f** Captured temperature distributions at steady state after the field evolutions at the peaks of the corner, edge, and bulk spectra as indicated by the red, blue, and black dashed lines in (**c**), and their locations are indicated by the dashed white lines in **d**–**f**.

the presence of net fluxes or flows of conserved quantities within the system. It also describes the responses of generalized fluxes (the net flux/flow) to the generalized forces (quantity gradients) based on Onsager reciprocal relations, which builds a common physical ground for extensive transport phenomena (Supplementary Note 4.4) with the interplay between energy and field motions.

## Non-Hermitian quadrupole topological phase induced by intrinsic non-Hermiticity

We demonstrate that such quadrupole topological phases can also be enabled by the intrinsic non-Hermiticities and captured along the imaginary-valued bands (Supplementary Eq. 12). Note that, such states in these imaginary-valued bands can be theoretically observed either in a skew-Hermitian thermal system without advections

$(\Omega_I = \Omega_{II} = 0)$ or a non-Hermitian heat transfer with advections possessing the same magnitudes and direction $(\Omega_I = \Omega_{II} \neq 0)$. Here, we focus on the non-Hermitian strategy and further demonstrate the quadrupole topological phases as illustrated in Fig. 3a $(\Omega_I = \Omega_{II} = 0.025Q_c)$. In that case, the real-valued band is gapless and can be adopted to distinguish the states along the gapped imaginary-valued band (Supplementary Note 4). The intracell and intercell thermal coupling strengths should be also different at this stage, since the same coupling strengths would otherwise close the imaginary-valued bands and indicate a trivial bulk state[24,25] instead. The coupling strengths can be manipulated by the heat exchanges within the intercell and intracell channels. For simplification, we keep the same intercell coupling channels as the case shown in Fig. 1. We further modify the structure by enlarging the intracell coupling

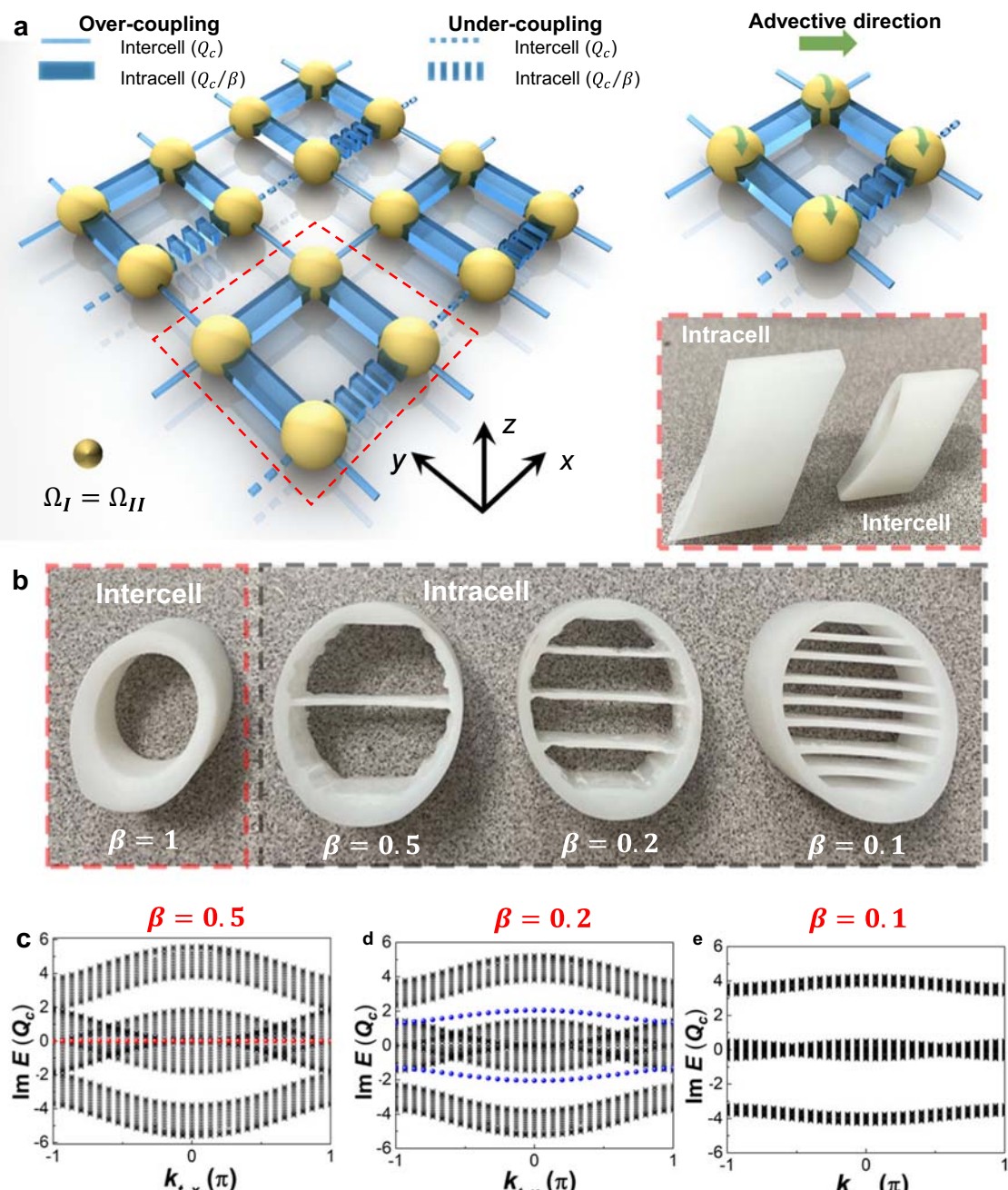

**Fig. 3 | Quadrupole topological phases in heat transport induced by non-Hermitian couplings and relevant band structures.** **a** illustrates the square-lattice with different coupling strengths and the four-site unit structure. The directions and magnitudes of the imposed advections (green arrows) on each site are same to hold the non-Hermitian properties. The intercell and intracell channels are fabricated to different structures to enable the different thermal coupling channels (the right lower inset of Fig. 3a) and inserting internal fins (Fig. 3b). Such implementations lead to stronger intracell thermal couplings under the same energy inputs and enable the modulations of $\beta$ ranging from 0 to 1. The imaginary band structures of the first Brillouin zones of one modified square-lattice are presented in Supplementary Fig. 6b. Similar to the modulations with Hermitian advection (Fig. 1c), all the imaginary-valued bands degenerate with the same intracell and intercell coupling strengths ($\beta = 1$). Two gaps (one between the first and second bands, and the other between third and fourth bands) are observed when modulating $\beta$, thus revealing the 1D edge and 0D corner states in the imaginary-valued bands. The strengths. **b** presents the inner structures of these thermal coupling channels with different coupling strength ratios. **c–e** indicate the imaginary spectra of the dispersion induced by non-Hermitian couplings respectively at the in-gap corner, gapped edge, and trivial bulk states. The horizontal and vertical axes denote the effective Bloch wave numbers and imaginary value of the eigenvalue. The red and blue dots in c and d respectively indicate the corner and edge states.

dispersion relations further validate the existences of in-gap corner (Fig. 3c), gapped edge (Fig. 3d), and trivial bulk (Fig. 3e) states along the imaginary-valued bands at tailored $\beta$.

We construct the thermal system with 9 modified square-lattices as illustrated in Fig. 4a and Supplementary Fig. 6c. When $\beta$ respectively approaches near-zero and 1 in the experiments, two branches are localized along Im$f$ and imply the trivial bulk states (Fig. 4b). When selecting $\beta$ in the range of 0 to 1, two gapped edge and one in-gap corner states are also expected. The field intensities on imaginary-valued bands (Fig. 4c) further verify the above hypothesis with two peaks on the central/edge and one peak on the corner of the

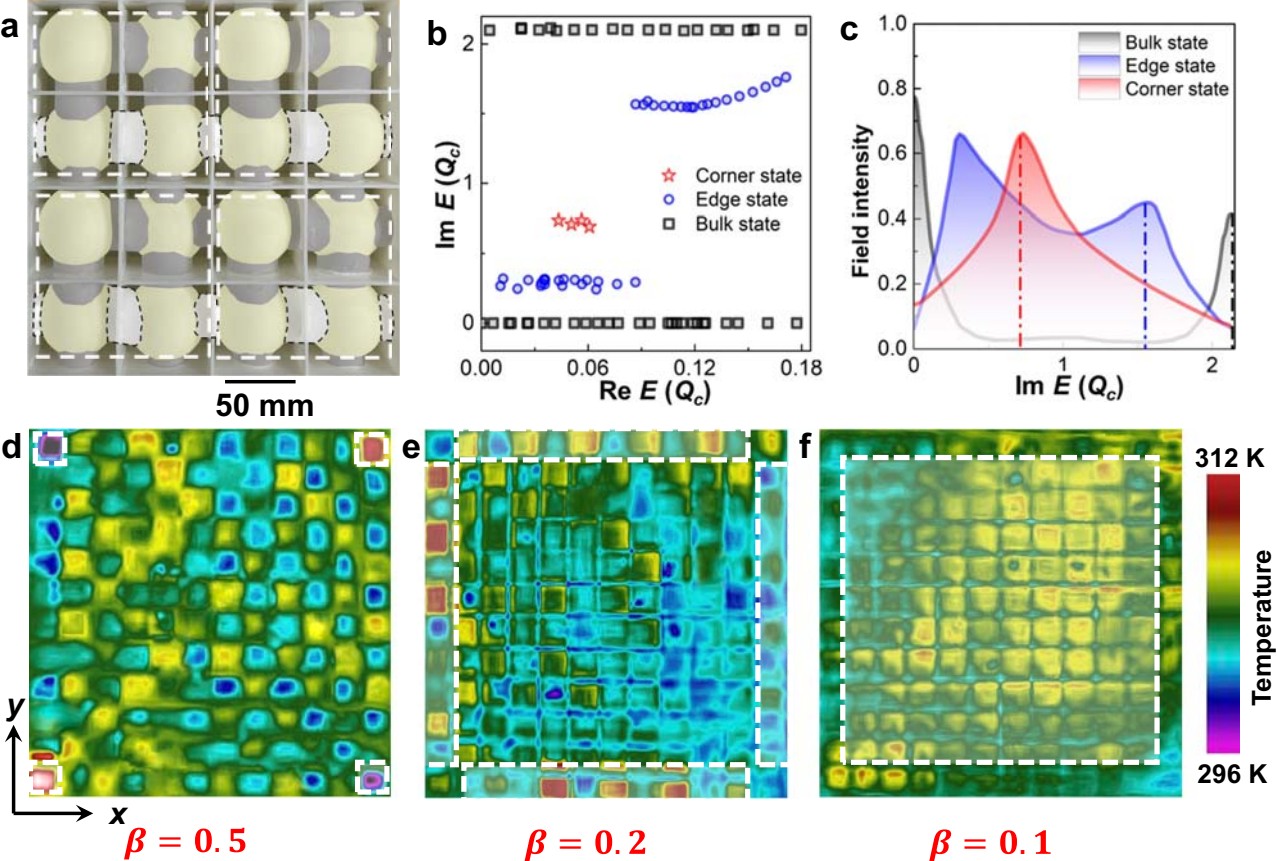

**Fig. 4 | Observation of the quadrupole topological phases in heat transport solely induced by non-Hermitian couplings. a** Photo of a fabricated square-lattice (16 sites) with different intercell and intracell coupling channels. The grey-shadowed and black-dashed areas respectively indicate the over-coupling and under-coupling channels. **b** Spectra of the thermal quadrupole topological phases induced by non-Hermitian couplings. **c** Measured temperature field intensities at measured sites (Supplementary Fig. 6c). Such features agree well with the numerical results (Supplementary Fig. 8) and experimental temperature field distributions (Fig. 4d–f). corresponding boundaries. The measured regions are marked by colored borders in Supplementary Fig. 6c. **d–f** Captured temperature distributions at steady state after the field evolutions at the peaks of the corner, edge, and bulk spectra as indicated by the red, blue, and black dashed lines in (**c**), and their locations are indicated by the dashed white lines in **d–f**. $\beta$ is the ratio between the intercell and intracell thermal coupling strengths.

measured sites (Supplementary Fig. 6c). Such features agree well with the numerical results (Supplementary Fig. 8) and experimental temperature field distributions (Fig. 4d–f). These results (Figs. 2 and 4) demonstrate the proof-of-concept quadrupole topological phases in non-Hermitian thermal systems via controlling either the imposed advections (Hermiticity) or the thermal coupling strengths (non-Hermiticity). Their topological robustness can be described by the nontrivial quadrupole invariant ($\frac{1}{2}$) and half-integer polarizations ($\frac{1}{2}$) based on the Wannier bands in the Brillouin zone and the nested Wilson loop respectively along the $x$ and $y$ directions in the parameter space of the fluid heat transport system (Supplementary Note 4). The calculated polarizations for the results in Figs. 2 and 4 indicate both the two strategies possess gapped Wannier bands and half-integer quantized polarizations (Supplementary Fig. 4). Moreover, these hierarchical states are also significant during none-quilibrium processes before reaching stable (Figs. 2 and 4), which can be described by the time changing rate of field intensity $\frac{\partial I}{\partial t}$ (Supplementary Note 5).

## Discussion

We report the creation of an effective quadrupole moment in heat transport and observe the non-Hermitian thermal quadrupole topological phases. Our results highlight the fundamental properties of these higher-order diffusive quadrupoles that drastically deviate from the wisdom about HOTIs in classical wave fields. The complex eigenvalues enable the phase transitions on both the real- and imaginary-valued bands. By modulating either the Hermitian advection or the non-Hermitian thermal coupling, the experimental demonstrations exhibit significant hierarchies of topological states in heat transport. Quadrupole topological phases in diffusive domains may reveal exotic physics on complex bands and empower the topological diffusion in fractal systems[39] and moiré lattices[40,41]. These diffusive bulk, edge and corner states as discovered in this work may further shed lights on the control of mass concentration in biomedicine and catalysis as well as the charge diffusion in semiconductors, and many other diffusive fields at large (Supplementary Note 6).

## Methods

### Experimental samples and coupling channel

We fabricated two types of experimental samples to demonstrate the non-Hermitian thermal quadrupole topological phases enabled by the advections and thermal couplings, as illustrated in Fig. 2 and Supplementary Fig. 6c, i.e., the advective and coupling types. All these fabricated samples consist of 144 sites with corresponding hollow advective balls shaped in the same radii of 25 mm, and 264 coupling channels for connecting any two neighboring sites. The wall thicknesses of these advective balls and coupling channels are 1 mm.

In order to hold the entire system within fluid ambient and independently implement the tailored modulations on each site, we set a series of square blocks possessing small thicknesses around each advective ball (Fig. 2 and Supplementary Fig. 6c). The entire system is installed via setting the advective balls in each block and embedding the coupling channels on the square partitions. Then, water is fully infused into each region, hollow advective ball, and coupling channel for creating the fluid ambient. The tailored advections of each type are modulated by independent motors through specific steering gears inside the advective balls (Supplementary Fig. 6d). Note that, the configurations of the advective and coupling types (samples) are completely different. For the sample modulated by advections, all the intracell and intercell channels are same to maintain the same thermal coupling strengths (Fig. 3b) to satisfy the condition of $\beta = 1$.

For the sample modulated by thermal coupling strength, we keep the same intercell coupling channels with the advective types and modify the intracell channels to reach the appropriate thermal coupling strength ratios under the same energy inputs. Based on the Newton cooling law, the total coupling energies within each channel are directly proportional to the heat exchange areas. In that case, more average temperature distributions are significant with larger heat exchange areas, and localized temperature occur with small areas. Thus, we can generally modulate them by changing related heat exchange areas (Fig. 3a, b). When $\beta = 0.5$, we increase the internal heat exchange areas within each intracell channel via simultaneously enlarging the entire channel size and inserting one internal fin to maintain the approximate thermal coupling strength ratio. For the case of $\beta = 0.2/0.1$, we further configure certain numbers of fins to each enlarged intracell channel to respectively realize the five/ten times the total heat exchange areas of the intercell ones (Fig. 3b).

### Systemic parameters, actuation, and general setups

The fabricated samples satisfy spatial periodicities both along $x$ and $y$ directions. Considering the square lattice adopted in the current system, the distances between neighboring centers of any two advective balls are designed as $a_x = a_y = a = 56$ mm. The internal width of each block for holding the advective ball and coupling channels is 56 mm. For creating the fluid ambient within the entire system, water with a thermal conductivity of 0.6 W·m$^{-1}$·K$^{-1}$ is adopted. To weaken the additional thermal effects between the sample and injected water, all the samples, advective balls, and coupling channels are made of epoxy resin, whose thermal conductivity is also 0.6 W·m$^{-1}$·K$^{-1}$ ($\rho = 1180$ kg·m$^{-3}$, $c = 750$ J·kg$^{-1}$·m$^{-3}$). For the actuation of these advective balls, we introduced a steering gear set to each advective ball (Supplementary Fig. 6d). Such a steering gear set consists of a pair of bevel gears (12 teeth, transmission ratio is 1), which is available to provide driving motions from different directions. Considering the superposed velocity fields respectively in the $x$-$z$ and $y$-$z$ planes, we can actuate the modulated angular velocities with independent motors via the transmission shafts along $z$-direction, and the steering gear sets further switch the motional directions. Such behaviors lead to the rotations around the axis perpendicular to $z$-direction and raise the effective advective components out of the $x$-$y$ plane. For manipulating the projections of the superposed velocity fields on $x$-$y$ plane, we only need to adjust the orientations of the advective balls to satisfy the specific demands.

During the experiments, the initial temperature profiles are imposed by hot (323 K) and cold (283 K) waters in corresponding blocks to satisfy the field distributions of effective thermal quadrupoles, while the ambient temperature is 297 K (right-inset of Supplementary Fig. 6c). It is noted that some deviations in the temperatures caused by the heat exchanges between blocks and ambient are observed due to the sequential orders of water injections (Supplementary Fig. 6e). Then, the motors are started to modulate the systems

at specific advections and coupling effects. All the temperature distributions are captured by an IR camera with a setting emissivity of 0.97. For simplifying the observations of measured intensities, the average temperatures $\bar{T}_{blc}$ of each block are adopted to replace $T^*$ used in the theoretical calculations via $I = |\frac{(\bar{T}_{blc} - \bar{T}_{mea})}{\Delta T_{mea}}|$. All the temperatures used for these calculations are directly measured by thermocouples.

### Experimental demonstrations for thermal quadrupole topological phases with Hermitian advection

Based on the advective demands, the critical strategy for observing the thermal quadrupole topological phases with Hermitian advection is to modulate the velocity differences between the imposed advections. Considering the structural parameters (the thicknesses of the advective balls and coupling channels) and thermal properties of the system (water and epoxy resin), the convective heat transfer coefficients of the intercell and intracell channels can be estimated with the Bartz equation, i.e., $h \rightarrow 5696$ W·m$^{-2}$·K$^{-1}$. In that case, the value of $Q_c$ is 0.129 s$^{-1}$ and $\Omega_I = 1.3Q_c = 0.0205$ rad·s$^{-1}$. The other angular velocities $\Omega_{II}$ of the cases shown in Supplementary Fig. S7g–i are respectively $\Omega_{II} = -1.8Q_c = -0.0283$ rad·s$^{-1}$, $\Omega_{II} = -2.7Q_c = -0.0424$ rad·s$^{-1}$, and $\Omega_{II} = -4.1Q_c = -0.0647$ rad·s$^{-1}$. The imposed velocities of each case adopted in the experiments strictly follow the above theoretical values, and the thermal profiles are captured when the changing trends of temperature distributions of each region become stable (about 30 min after activating the motors).

### Experimental demonstrations for thermal quadrupole topological phases with non-Hermitian thermal couplings

In the experimental demonstrations with non-Hermitian thermal couplings, the imposed advections are only used for providing the Hermiticities rather than modifying the effective bands. Thus, we adopt the same and quite small angular velocities for the advective configurations ($\Omega_I = \Omega_{II} = 0.025Q_c = 0.0004$ rad·s$^{-1}$). As mentioned above, three kinds of intracell coupling channels are fabricated to satisfy the tailored thermal coupling strength ratios of the cases shown in Figs. 3, 4, and Supplementary Fig. 6. Thus, three independently sub-demonstrations are implemented via switching different intracell coupling channels in turn. During the measurements, it takes a longer time to reach the stable state (about 50-60 min after activating the motors) than that with Hermitian advection, since the heat exchange components induced by the advections are far smaller than the ones of intrinsic conductions.

## Data availability

The data supporting the findings of this study are available within the article and its supplementary file. Data for the figures can be found in the file of Source Data. Source data are provided with this paper.

## Code availability

The code utilized during the current study is available from the corresponding author on request.

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

## Acknowledgements

C.-W.Q. acknowledged the financial support by Ministry of Education, Republic of Singapore (Grant No.: A-8000107-01-00) and the National Research Foundation, Singapore (NRF) under NRF's Medium Sized Centre: Singapore Hybrid-Integrated Next-Generation μ-Electronics (SHINE) Centre funding programme. X.Z. acknowledged the financial support of Chongqing Natural Science Foundation (Grant No. cstc2021jcyj-msxmX0627) and the Science and Technology Research Program of Chongqing Municipal Education Commission (Grant No. KJQN202000829). J.W. acknowledges the SERC Central Research Fund and Advanced Manufacturing and Engineering Young Individual Research Grant (AME YIRG Grant No.: A2084c170).

## Author contributions

G.X. and C.-W.Q. conceived the idea. G.X., X.Z., and C.-W.Q. proposed the methodology. G.X. and S.H.Y. performed the theoretical derivation, and G.X., and X.Z. implemented the experimental investigations. G.X., X.Z., S.H.Y., J.W., and C.-W.Q. made the visualizations. G.X., S.H.Y., J.W., and C.-W.Q. performed the theoretical analysis and wrote the manuscript. C.-W.Q. supervised the work. All authors contributed to the discussion and finalization of the manuscript.

## Competing interests

The authors declare no competing interests.
