## [Peer Review File · Nature Communications]

REVIEWER COMMENTS

Reviewer #1 (Remarks to the Author):

This study investigates non-Hermitian quadrupole topological insulators and their corresponding higher-order features in fluid heat transport, using both experimental and theoretical approaches. By utilizing fluid dynamics and specialized configurations of fluid channels, the authors are able to achieve critical over-coupling and under-coupling components in the fluid heat transport. They further observe hierarchical topological features in both the real and imaginary bands, respectively controlling Hermitian advection (Hermiticity) and thermal coupling strengths (non-Hermiticity). The authors provide detailed discussions of these topological features using Wannier bands and the symmetry of the lattice. In addition, the authors discover that these topological features can emerge simultaneously in multiple fields of fluid dynamics due to the similar essences of the transported elements within the fluid, such as heat, mass, and momentum, governed by conservation laws.

This study is both fascinating and well-written. It offers comprehensive demonstrations and opens up new avenues for exploring non-Hermitian higher-order topology based on a continuous model in fluid heat transport. This approach differs from the state-of-the-art topology in wave dynamics and pure diffusion, which are respectively governed by purely real and imaginary couplings. By adopting fluid dynamics, this work extends to multiple diffusive and momentum fields within the fluid transport processes. This is a noteworthy addition that is lacking in conventional topological studies related to wave dynamics. Understanding the topological mechanisms behind the hierarchical behaviors in fluid heat transport also enriches conventional thermal physics and techniques with many applications that can address various heat management and thermoelectric problems. Therefore, this work is of interest to a broad audience, and I recommend its publication in *Nat. Commun.* I have only a few minor comments that should be addressed before publication.

1. The authors have made a detailed discussion of the topological robustness of these observed behaviors with Wannier bands and nested Wannier bands in the Supplementary Information. These topological invariants are critical indicators for the nontrivial behaviors in quadrupole topological phases. Some brief illustrations about these topological invariants should be mentioned in the main content.

2. The authors have simultaneously presented the topological behaviors in multiple transport fields within the fluid dynamics in the Supplementary Information. This is a quite strong and instructive point for the topological physics in dissipative transports. This is due to the same physical essence under conservation laws and similar constitutive equations for different transport phenomena. I suggest adding some brief illustration about this point to the main content.

3. The authors have illustrated the thermalizing strategy for the experiments with the alternative configuration of hot and cold water in neighboring sites in Method, and further provided an initial temperature field in Supplementary Figure 6e. For a better presentation and understanding of the experimental setup, a schematic subgraph of such an alternative configuration in one four-unit structure can be made as a representation.

4. I would suggest replacing the phrase “phase diagram” with “band structure” in both the main text and supplementary information, as “band structure” is a more standard term in condensed matter physics.

5. In the paragraph beginning with “In Supplementary Equation 3” on page 3 of the supplementary information, there appears to be a typographical error, as two extra “h”s are present in the relations of heat flux components. This should be corrected.

6. In both the main text and supplementary information, there are multiple instances of the words “titled” and “tilted”. I believe that all instances of “titled” should be corrected to “tilted”, as this appears to be a clerical error.

7. In the caption of Supplementary Figure 6, I suggest revising the first sentence to read “Band structures of the first Brillouin zones induced by Hermitian advections and non-Hermitian thermal couplings” instead of “Phase diagrams of the first Brillouin zones induced by non-Hermitian thermal couplings”, as this better reflects the content of Supplementary Figure 6a.

8. There have been several theoretical and experimental proposals put forward regarding the realization of higher-order topological insulators in diffusion systems [e.g., Liu et al., arXiv:2206.09837 (2022); Wu et al., Adv. Mater. 202210825 (2023)].

Reviewer #2 (Remarks to the Author):

This manuscript develops a new method for creating quantized bulk quadrupole moments in fluid heat transport, which allows for the observation of quadrupole topological phases in non-Hermitian thermal systems. The authors observe hierarchical hallmarks of bulk, gapped edge, and in-gap corner states in both the real- and imaginary-valued bands in non-Hermitian thermal systems, which is in contrast to the

higher-order states that are only observed on real-valued bands in classic wave fields. This work represents an advance in the field of thermal metamaterials, as it demonstrates the potential for new discoveries and technologies based on the manipulation of thermal transport. It also establishes a new playground for multipolar topological physics, which has implications for other areas of physics and engineering.

Personally, I find this work quite exciting and I would recommend it to be published on Nat. Com. Nevertheless, a few questions need to be addressed first:

1. In supplementary figure 13, the manuscript has demonstrated the robustness of corner and edge states by putting a defect at the lower right corner. The behavior of the edge states is easy to understand, and I'm more interested in the corner state. In particular, I believe there should be some quantity conserved in the field to make the corner state robust. Have you found that quantity? Have you checked other configurations to make sure it's a conserved quantity?
2. In Supplementary Note 6, the manuscript has claimed that the localized position can be anywhere in the system. I'm wondering whether there's any evidence to support this argument. If not, how has this argument arrived? If so, under what condition can it be observed? For example, does the lattice translational symmetry need to be broken?
3. A few minor issues. Headings can be helpful in the main article to help readers follow the logic. The color bar should be added to Supplementary Figure 1cd. Also, there's a typo in the subscript of Ω in Supplementary Equation 1.

**Response to the Reviewers' Comments on the manuscript [NCOMMS-23-12653] entitled
"Observation of bulk quadrupole in topological heat transport" submitted to *Nature
Communications***

We would like to thank the editors and reviewers for their careful reviewing of our work and their relevant comments. These comments are quite valuable, based on which we further improve the descriptions of our work. We have made explicit illustrations of the role of the topological invariants and the conserved quantities. We have further provided some examples of the robustness of the topological corner states and the evidence regarding the localization of heat energies at arbitrary positions in the system. The changes in the revised manuscript are marked in red. In what follows, we make a point-to-point response to all reviewers' comments. We hope our efforts may provide sufficient illustrations on all the concerns, and the current version may satisfy all the critical criteria of *Nature Communications*.

Reviewer #1 (Remarks to the Author):

This study investigates non-Hermitian quadrupole topological insulators and their corresponding higher-order features in fluid heat transport, using both experimental and theoretical approaches. By utilizing fluid dynamics and specialized configurations of fluid channels, the authors are able to achieve critical over-coupling and under-coupling components in the fluid heat transport. They further observe hierarchical topological features in both the real and imaginary bands, respectively controlling Hermitian advection (Hermiticity) and thermal coupling strengths (non-Hermiticity). The authors provide detailed discussions of these topological features using Wannier bands and the symmetry of the lattice. In addition, the authors discover that these topological features can emerge simultaneously in multiple fields of fluid dynamics due to the similar essences of the transported elements within the fluid, such as heat, mass, and momentum, governed by conservation laws.

Response: We are thankful to the reviewer for providing the accurate summaries of our work.

This study is both fascinating and well-written. It offers comprehensive demonstrations and opens up new avenues for exploring non-Hermitian higher-order topology based on a continuous model in fluid heat transport. This approach differs from the state-of-the-art topology in wave dynamics and pure diffusion, which are respectively governed by purely real and imaginary couplings. By adopting fluid dynamics, this work extends to multiple diffusive and momentum fields within the fluid transport processes. This is a noteworthy addition that is lacking in conventional topological studies related to wave dynamics. Understanding the topological mechanisms behind the hierarchical behaviors in fluid heat transport also enriches conventional thermal physics and techniques with many applications that can address various heat management and thermoelectric problems. Therefore, this work is of interest to a broad audience, and I recommend its publication in *Nat. Commun.* I have only a few minor comments that should be addressed before publication.

Response: We are thankful to the reviewer for recognizing our proposed paradigm toward realizing the bulk quadrupole in non-Hermitian fluid heat transport, the potentials in a plethora of fields in science and engineering, the endorsement on the significances of our current work, and the positive recommendation. We appreciate the related comments on our manuscript. We would like to make some responses to the constructive comments in the following content to further improve the current work.

1. The authors have made a detailed discussion of the topological robustness of these observed behaviors with Wannier bands and nested Wannier bands in the Supplementary Information. These topological invariants are critical indicators for the nontrivial behaviors in quadrupole topological phases. Some brief illustrations about these topological invariants should be mentioned in the main content.

Response: We are thankful to the reviewer for providing this comment. For a quadrupole topological insulator, the bulk topological quadrupole moment generates a topologically polarized edge when the system is cut parallel to either the x or y directions. This is a critical property for quadrupole topological insulators and the half-integer polarizations based on nested Wilson loop can be also considered as the high-order topological invariant for describing the quadrupole topological phase. As indicated by the reviewer, we have provided detailed discussions about the Wannier bands and nested Wannier bands in Supplementary Note 4 of the Supplementary Information. To further emphasize this point, we have made some illustrations in the main content. The related revision is as follow.

In the main content:

“Their topological robustness can be described by the nontrivial quadrupole invariant ($\frac{1}{2}$) and half-integer polarizations ($\frac{1}{2}$) based on the Wannier bands in the Brillouin zone and the nested Wilson loop respectively along the x and y directions in the parameter space of the fluid heat transport system (Supplementary Note 4). The calculated polarizations for the results in Figs. 2 and 4 indicate both the two strategies possess gapped Wannier bands and half-integer quantized polarizations (Supplementary Figure 4).”

2. The authors have simultaneously presented the topological behaviors in multiple transport fields within the fluid dynamics in the Supplementary Information. This is a quite strong and instructive point for the topological physics in dissipative transports. This is due to the same physical essence under conservation laws and similar constitutive equations for different transport phenomena. I suggest adding some brief illustration about this point to the main content.

Response: We are thankful to the reviewer for providing this thoughtful comment, and we also appreciate the reviewer on pointing out the intrinsicity of these topological behaviors in multiple transport fields, i.e., the same physical essence under conservation laws and similar constitutive equations for different transport phenomena. We have made some illustrations in Supplementary Note 6. To further emphasize this point and make some inspirations, we have added the following illustrations in the main content.

In the main content:

“All these demonstrated fields are typical transport phenomena within the fluids. These transport quantities of energy, mass, and momentum follows the conservation laws formulated by the constitutive equations of continuity equation, momentum equation, and energy equation of the fluid heat transport. The same continuous mechanism and mathematical frameworks of the constitutive equations between these different fields enable the similar processes of conserved transport and quantization, which can be generally described by the balance among the conserved quantities entering and leaving the control volume, the additional generations of all the original components for the conserved quantities, and the non-zero accumulations (the net flows) retaining in the system after the conservation process. Such a process is characterized by the presence of net fluxes or flows of conserved quantities within the system. It also describes the responses of generalized fluxes (the net flux/flow) to the generalized forces (quantity gradients) based on Onsager reciprocal relations, which builds a common physical ground for extensive transport phenomena (Supplementary Note 4.4) with the interplay between energy and field motions.”

3. The authors have illustrated the thermalizing strategy for the experiments with the alternative configuration of hot and cold water in neighboring sites in Method, and further provided an initial temperature field in Supplementary Figure 6e. For a better presentation and understanding of the experimental setup, a schematic subgraph of such an alternative configuration in one four-unit structure can be made as a representation.

Response: We thank the reviewer for providing this comment. A schematic subgraph of such a thermalizing strategy in one four-unit structure has been added in Supplementary Figure 6.

In the Supplementary Information (Supplementary Figure 6):

Supplementary Figure 6. Band structures of the first Brillouin zones induced by Hermitian advectives and non-Hermitian thermal couplings, fabricated samples in Fig. 4, and the initial thermal profile without advectives. a denotes the real-valued **band structures** of the first Brillouin zones under tailored advectives. **b** presents the imaginary **band structures** of the first Brillouin zones under tailored β . The flip of the “+” and “-” signs in **(a)** and **(b)** implies the topological phase transition with the changing advectives and β . **c**. Experimental sample for observing the quadrupole topological phases induced by non-Hermitian thermal couplings (Fig. 4 of the main content). **The right insert denotes the schematic subgraph of the thermalizing strategy in one four-unit structure with alternative hot and cold waters. d**. The steering gear set consisting of a pair of bevel gears for modulating the motions of each advective ball. **e**. The initial temperature distribution without advectives.”

4. I would suggest replacing the phrase “phase diagram” with “band structure” in both the main text and supplementary information, as “band structure” is a more standard term in condensed matter physics.

Response: We are thankful to the reviewer for providing this suggestion. The related expressions have been revised.

In the main content:

“**Fig. 1.** Quadrupole topological phases in heat transport induced by Hermitian advection and **band structure.**”

“**Fig. 3.** Quadrupole topological phases in heat transport induced by non-Hermitian couplings and **band structure.**”

In the Supplementary Information:

“The imaginary **band structures** of the first Brillouin zones under specific β are shown in Supplementary Figure 6b.”

“The real **band structures** of the first Brillouin zones under specific advectons are presented in Supplementary Figure 6a.”

“**Supplementary Figure 6.** **Band structures of the first Brillouin zones induced by Hermitian advectons and non-Hermitian thermal couplings**, fabricated samples in Fig. 4, and the initial thermal profile without advectons. **a** denotes the real-valued **band structures** of the first Brillouin zones under tailored advectons. **b** presents the imaginary **band structures** of the first Brillouin zones under tailored β .”

5. In the paragraph beginning with “In Supplementary Equation 3” on page 3 of the supplementary information, there appears to be a typographical error, as two extra “h”s are present in the relations of heat flux components. This should be corrected.

Response: We thank the reviewer for the careful reading. The typos have been revised accordingly.

In the Supplementary Information:

$$“ q_{ij \sim i+1j}(\mathbf{x}) = h \left(T_{i+1j}(\mathbf{x}) - T_{ij}(\mathbf{x}) \right) = \beta e^{ik_t x \cdot 4a} q_{i-1j \sim ij}(\mathbf{x}) \quad \text{and} \quad q_{ij+2 \sim ij+3}(\mathbf{y}) = h \left(T_{ij+3}(\mathbf{y}) - T_{ij+2}(\mathbf{y}) \right) = \beta e^{ik_t y \cdot 4a} q_{ij+1 \sim ij+2}(\mathbf{y}). ”$$

6. In both the main text and supplementary information, there are multiple instances of the words “titled” and “tilted”. I believe that all instances of “titled” should be corrected to “tilted”, as this appears to be a clerical error.

Response: We thank the reviewer for the careful reading. The typos have been revised accordingly.

In the main content:

“The **tilted** connections (Supplementary Figure 2) between two adjacent sites result in different orientations of isotherms and coupling degrees under the same advectons (Supplementary Figure 3).”

In the Supplementary Information:

“Here, we design two types of **tilted** connections to create different thermal coupling orientations (Supplementary Figure 2).”

“**Supplementary Figure 3.** Temperature distributions of the intracell channels with **tilted** connections.”

7. In the caption of Supplementary Figure 6, I suggest revising the first sentence to read “Band structures of the first Brillouin zones induced by Hermitian advectons and non-Hermitian thermal couplings” instead of “Phase diagrams of the first Brillouin zones induced by non-Hermitian thermal couplings”, as this better reflects the content of Supplementary Figure 6a.

Response: We thank the reviewer for the suggestion. The caption has been revised accordingly.

In the Supplementary Information:

“**Supplementary Figure 6.** Band structures of the first Brillouin zones induced by Hermitian advectons and non-Hermitian thermal couplings,”

8. There have been several theoretical and experimental proposals put forward regarding the realization of higher-order topological insulators in diffusion systems [e.g., Liu et al., arXiv:2206.09837 (2022); Wu et al., *Adv. Mater.* 202210825 (2023)].

Response: We thank the reviewer for the providing these two references related to higher-order topology in purely thermal conduction. We have mentioned them in the revision.

In the main content:

“36. Liu, Z., Xu, L., Huang, J. Higher-dimensional topological insulators in pure diffusion systems. arXiv:2206.09837 (2022).

37. Wu, H., Hu, H., Wang, X., Xu, Z., Zhang, B., Wang, Q. J., Zheng, Y., Zhang, J., Cui, T. J., Luo Y. Higher-order topological states in thermal diffusion. *Adv. Mater.* **35**, 202210825 (2023).”

Reviewer #2 (Remarks to the Author):

This manuscript develops a new method for creating quantized bulk quadrupole moments in fluid heat transport, which allows for the observation of quadrupole topological phases in non-Hermitian thermal systems. The authors observe hierarchical hallmarks of bulk, gapped edge, and in-gap corner states in both the real- and imaginary-valued bands in non-Hermitian thermal systems, which is in contrast to the higher-order states that are only observed on real-valued bands in classic wave fields. This work represents an advance in the field of thermal metamaterials, as it demonstrates the potential for new discoveries and technologies based on the manipulation of thermal transport. It also establishes a new playground for multipolar topological physics, which has implications for other areas of physics and engineering.

Response: We are thankful to the reviewer for the carefully reading and the accurate summaries of our work.

Personally, I find this work quite exciting and I would recommend it to be published on Nat. Com. Nevertheless, a few questions need to be addressed first:

Response: We are thankful to the reviewer for the positive recommendation, and we also appreciate the related comments on our manuscript. We would like to make some responses to the constructive comments in the following content to further improve the current work.

1. In supplementary figure 13, the manuscript has demonstrated the robustness of corner and edge states by putting a defect at the lower right corner. The behavior of the edge states is easy to understand, and I'm more interested in the corner state. In particular, I believe there should be some quantity conserved in the field to make the corner state robust. Have you found that quantity? Have you checked other configurations to make sure it's a conserved quantity?

Response: We thank the reviewer for providing this thoughtful comment. The robustness of these corner states shown in Supplementary Figure 13 is enabled by a fractional charge based on the local density of state (LDOS) (Refs. [22] and [39], *Sci. Bull.* **67**, 2040-2044, (2022), and *Sci. Bull.* **67**, 2069-2075, (2022)). In these cases (Supplementary Figure 13), defects are imposed by removing one entire square-lattice with 16 sites (right-upper insert of Fig. 2a and Fig. 4a) at the lower-right corner to create the imperfections in the sample geometry of the system. That is, the C_4 symmetry of the square-lattice remains but the translational symmetry fails within the system under such defects.

To illustrate this point, we calculate the sums of local density of state (LDOS) for these cases shown in Supplementary Figure 13a and d under such defects (Fig. R1). Both the findings for the cases induced by Hermitian advections (Fig. R1a) and non-Hermitian thermal couplings (Fig. R1b) indicate that the corners without defects possess half-integer quantized invariant (0.5), which is an important characteristic of the higher-order quadrupole insulator with quantized quadrupole moment based on Wannier bands and nested Wannier bands (Supplementary Note 4). Different from the above regular corner states (without defects) spanning over an angle of $\frac{\pi}{2}$, it also presents that the two neighboring sites of the interior corner, which further generate a trimer, exhibit the fractional charges of 0.25. This

is caused by the $\frac{3\pi}{2}$ spanning over the corner with defects, which split the charge of 0.5 into two halves at the neighboring sites. Such fractional charges of 0.25 are also the significant indicator for describing the robust corner states with fractional charges (Refs. [22, 39], *Sci. Bull.* **67**, 2040-2044, (2022), and *Sci. Bull.* **67**, 2069-2075, (2022)).

Such a fractional charge (conserved quantity) is found in the wavefunction of the fluid transport system. Thus, the corresponding states can be also observed in different configurations based on the conservation laws for the fields of energy, mass, and momentum within the fluid, whose transport properties satisfy the wavefunction and further result in the non-zero accumulations and net flow in related fields at these corners. The related configurations in other fields are shown in Figs. R1 **c ~ f**. Note that, the pressure and velocity fields are quite small in the case induced by non-Hermitian couplings (Fig. R1 **f**), since the near-zero and same advectons are imposed on each site ($\Omega_I = \Omega_{II} = 0.025Q_c$ mentioned in the main content), and lead the transport process within the entire system to quasi-conduction with little effects of advectons. These robust corner states in different transport field within the fluid further indicate the significance of the conserved quantity (fractional charge).

Fig. R1. Topological charge and corner states for different fields under the defects of imperfections in the sample geometry. **a** and **b** respectively denote the topological charges (quantized quadrupole invariant) for the cases induced by Hermitian advection and non-Hermitian couplings. Both these cases showcase fractional charges of 0.5 at the regular corner without defects and 0.25 at the neighboring sites of the interior corner at the sample defects. **c ~ e** present the momentum fields (pressure and velocity) and mass concentration field for the cases induced by Hermitian advectons (the imposed advectons are same with the cases shown in Fig. 2a of the main content). **f** plots the mass concentration field induced by non-Hermitian coupling ($\beta = 0.5$) under near-zero and same advectons (quasi-conduction system). All these field distributions indicate significant corner states protected by these topological charges.

To further make such a point clear, we have added the above illustrations and Fig. R1 to Supplementary Note 4.6. The related revisions are as below:

In Supplementary Note 4.6:

“The robustness of these corner states shown in Supplementary Figures 13a and d is enabled by a fractional charge based on the local density of state (LDOS) in Refs. [22, 39] of the main content. In these cases, defects are imposed by removing one entire square-lattice with 16 sites (right-upper insert of Fig. 2a and Fig. 4a) at the lower right corner to create the imperfections in the sample geometry of the system. That is, the C_4 symmetry of the square-lattice remains but the translational symmetry fails within the system under such defects. The sums of local density of state (LDOS) for these cases are presented in Supplementary Figures 14a and b. Both the findings for the cases induced by Hermitian advections (Supplementary Figures 14a) and non-Hermitian thermal couplings (Supplementary Figures 14b) indicate that the corners without defects possess half-integer quantized invariant (0.5), which is an important characteristic of the higher-order quadrupole insulator with quantized quadrupole moment based on Wannier bands and nested Wannier bands (Supplementary Note 4.1). Different from the above regular corner states (without defects) spanning over an angle of $\frac{\pi}{2}$, it also presents that the two neighboring sites of the interior corner, which further generate a trimer, exhibit the fractional charges of 0.25. This is caused by the $\frac{3\pi}{2}$ spanning over the corner with defects, which split the charge of 0.5 into two halves at the neighboring sites. Such fractional charges of 0.25 are also the significant indicator for describing the robustness corner states with fractional charges (Refs. [22, 39]). Such a fractional charge (conserved quantity) is found in the wavefunction of the fluid transport system. Thus, the corresponding states can be also observed in different configurations based on the conservation laws for the fields of energy, mass, and momentum within the fluid, whose transport properties satisfy the wavefunction and further result in the non-zero accumulations and net flow in related fields at the corners. The robust corner states in other field configurations within the fluid (Supplementary Figures 14c ~ f) further indicate the significance of the conserved quantity (fractional charge).”

2. In Supplementary Note 6, the manuscript has claimed that the localized position can be anywhere in the system. I'm wondering whether there's any evidence to support this argument. If not, how has this argument arrived? If so, under what condition can it be observed? For example, does the lattice translational symmetry need to be broken?

Response: We thank the reviewer for providing this thoughtful question. At the beginning, we would like to indicate that such a behavior is found in the prior studies of this work. Thus, we have made some corresponding outlooks in Supplementary Note 6. **Such behaviors can be generally realized by creating an internal bound state within the system, which break the lattice translational symmetry.** As an example, some state-of-art works exploit the butting of topological nontrivial and trivial lattices to form “long-long defects” at their boundaries (*Phys. Rev. Lett.* **121**, 213902 (2018)). Such a method requires the combinations of nontrivial and trivial lattice within the entire system, thus naturally break the translational symmetry at their boundaries. In our prior studies related to the current work, we found that the bound states can be also realized in one lattice type (such as the square-lattice of the current work) by inserting corresponding domain walls at tailored sites. In that case, the neighboring lattices at the interface are mismatched thus giving rise to the internal boundary states. Besides, the combinations of different lattices (*Phys. Rev. Lett.* **121**, 213902 (2018)) can be avoided to simplify the sample.

We provide two inner corner states and two inner edge states by inserting domain walls at tailored positions (Fig. R2). Note that, both the changes of Hermitian advection and non-Hermitian couplings

can result in the domain walls. In these illustrations, we maintain the advectations and couplings for the bulk states in corresponding cases (Fig. 2f and 4f of the main content). Then, the related domain walls can be created by reducing the on-site advectations (coupling-channel areas) to 0.01 time to the original one for the case induced by Hermitian advectations (non-Hermitian couplings). The findings imply that expected bound states with different quantities and positions are observed with these domain walls, thus enabling the energy localizations at arbitrary positions.

Fig. R2. Bound states in arbitrary positions of the system with domain walls. **a ~ d** denote the sites where domain walls are set within the system (black dots). The grey shadow indicates one square lattice. The domain walls are created by reducing related advectations or coupling-channel areas to 0.01 times to the original cases induced by Hermitian advectations or non-Hermitian couplings. **e ~ h** plot the temperature fields of the internal bound states (inner-corner and inner-edge states) with different quantities and positions induced by the Hermitian advectations. **i ~ l** present the temperature fields of the corresponding internal bound states induced by the non-Hermitian couplings.

We believe that such field localizations with domain walls deserve more explorations and attentions in the studies of thermal management and the applications of topological heat transport, and we also have an on-going work directly related to these behaviors. Thus, we mentioned this point in the outlook (Supplementary Note 6). To make the statement clear and provide some inspirations for further studies and practical applications, we have made some brief illustrations to indicate the mechanism. The related revision is as below:

In Supplementary Note 6:

“Further exploring the merging states, the localized positions can be anywhere in the system, which is not limited to the bulk, edge, and corners. Such behaviors can be generally realized by creating an internal bound state within the system¹, which break the lattice translational symmetry. As an example, some state-of-art works exploit the butting of topological nontrivial and trivial lattices to form “long-long defects” at their boundaries¹. Such a method requires the combinations of nontrivial and trivial lattice within the entire system, thus naturally break the translational symmetry at their boundaries. Another potential approach is to insert domain walls at tailored sites (Supplementary Figure 16) in the system possessing one lattice type (such as the square-lattice of the current work). In that case, the neighboring lattices at the interface are mismatched thus giving rise to the internal bound states. Such a property deserves more studies, and it is quite important for exploiting paradigm-shift thermal management in electronic devices and other industrial applications.”

3. A few minor issues. Headings can be helpful in the main article to help readers follow the logic. The color bar should be added to Supplementary Figure 1cd. Also, there's a typo in the subscript of Omega in Supplementary Equation 1.

Response: We thank the reviewer for indicating the suggestions and comments. The related revisions have been made in the revised main content and Supplementary Information.

In the main content (Headings for corresponding sections):

“**Introduction**”

“**Generation of non-Hermitian quadrupole topological insulator in fluid heat transport**”

“**Non-Hermitian quadrupole topological phase induced by Hermitian advection**”

“**Non-Hermitian quadrupole topological phase induced by intrinsic non-hermiticity**”

In Supplementary Equation 1:

$$\frac{\partial T_{ij}}{\partial t} = \sum_i^4 \left(\frac{\kappa}{\rho c} \frac{\partial^2 T_{ij}}{\partial \mathbf{x}^2} + \frac{\partial (\Omega_i \cdot R(\mathbf{x}) \cdot T_{ij})}{\partial \mathbf{x}} \right) + \sum_j^4 \left(\frac{\kappa}{\rho c} \frac{\partial^2 T_{ij}}{\partial \mathbf{y}^2} + \frac{\partial (\Omega_j \cdot R(\mathbf{y}) \cdot T_{ij})}{\partial \mathbf{y}} \right). \quad (1)$$

In Supplementary Figures 1c and d:

REVIEWERS' COMMENTS

Reviewer #1 (Remarks to the Author):

The present authors report bulk quadrupoles in topological heat transport, as an interesting extension of thermal dipoles experimentally realized. It's a good research indeed. The authors answered my questions satisfactorily. I think the revised version of the paper can be accepted as is.

Reviewer #2 (Remarks to the Author):

The authors have satisfied my prior concerns with either clarification, additional references, or plots, and I am satisfied with the revised manuscript.

**Response to the Reviewers' Comments on the manuscript [NCOMMS-23-12653A] entitled
"Observation of bulk quadrupole in topological heat transport" submitted to *Nature
Communications***

We would like to thank the editors and reviewers for their careful reviewing of our work and their relevant comments. These comments are quite valuable, based on which we further improve the descriptions of our work. We have made explicit illustrations of the editor's concerns and made the corresponding revisions. The changes in the revised manuscript are marked in **red**. We hope our efforts may provide sufficient illustrations on all the concerns, and the current version may satisfy all the critical criteria of *Nature Communications*.

Reviewer #1 (Remarks to the Author):

The present authors report bulk quadrupoles in topological heat transport, as an interesting extension of thermal dipoles experimentally realized. It's a good research indeed. The authors answered my questions satisfactorily. I think the revised version of the paper can be accepted as is.

Response: We thank the reviewer for the efforts on our work and the positive recommendation.

Reviewer #2 (Remarks to the Author):

The authors have satisfied my prior concerns with either clarification, additional references, or plots, and I am satisfied with the revised manuscript.

Response: We thank the reviewer for the efforts on our work and the positive recommendation.